# Effect of milk protein and whey permeate in large quantity lipid-based nutrient supplement on linear growth and body composition among stunted children: A randomized 2 × 2 factorial trial in Uganda

Joseph Mbabazi[1,2], Hannah Pesu[1], Rolland Mutumba[1,2], Suzanne Filteau[3], Jack I. Lewis[1], Jonathan C. Wells[4], Mette F. Olsen[1,5], André Briend[1,6], Kim F. Michaelsen[1], Christian Mølgaard[1], Christian Ritz[7], Nicolette Nabukeera-Barungi[2], Ezekiel Mupere[2☯], Henrik Friis[1☯], Benedikte Grenov[1☯]*

1 Department of Nutrition, Exercise and Sports, University of Copenhagen, Copenhagen, Denmark, 2 Department of Paediatrics and Child Health, Makerere University, Kampala, Uganda, 3 Department of Population Health, London School of Hygiene and Tropical Medicine, London, United Kingdom, 4 Childhood Nutrition Research Centre, Population Policy and Practice Research and Teaching Department, UCL Great Ormond Street Institute of Child Health, London, United Kingdom, 5 Department of Infectious Diseases, Rigshospitalet, Copenhagen, Denmark, 6 Tampere Center for Child, Adolescent and Maternal Health Research, Faculty of Medicine and Health Technology, Tampere University and Tampere University Hospital, Tampere, Finland, 7 The National Institute of Public Health, University of Southern Denmark, Copenhagen, Denmark

☯ These authors contributed equally to this work.
* bgr@nexs.ku.dk

## Abstract

### Background

Despite possible benefits for growth, milk is costly to include in foods for undernourished children. Furthermore, the relative effects of different milk components, milk protein (MP), and whey permeate (WP) are unclear. We aimed to assess the effects of MP and WP in lipid-based nutrient supplement (LNS), and of LNS itself, on linear growth and body composition among stunted children.

### Methods and findings

We performed a randomized, double-blind, 2 × 2 factorial trial among 12 to 59 months old stunted children in Uganda. Children were randomized to 4 formulations of LNS with MP or soy protein isolate and WP or maltodextrin (100 g/day for 12 weeks) or no supplementation. Investigators and outcome assessors were blinded; however, participants were only blinded to the ingredients in LNS. Data were analyzed based on intention-to-treat (ITT) using linear mixed-effects models adjusted for age, sex, season, and site. Primary outcomes were change in height and knee-heel length, and secondary outcomes included body composition by bioimpedance analysis (ISRCTN13093195). Between February and September 2020, we enrolled 750 children with a median age of 30 (interquartile range 23 to 41) months, with

**Data Availability Statement:** The Ugandan act on Data Protection and Privacy and the European act on General Data Protection Regulation do not allow for personal data to be made available to other researchers without prior written approval from

relevant institutions and authorities. The Data Protection Officer of the University of Copenhagen can be contacted about data inquiries at dpo@adm. ku.dk.

**Funding:** The trial was funded by Arla Food for Health, https://arlafoodforhealth.com/ (HF). Arla Food for Health, is a public-private research partnership between the University of Copenhagen, Aarhus University and the dairy company, Arla. Additional funds were obtained from the Danish Dairy Research Foundation, https:// danishdairyboard.dk/research/ddrf/ (HF), Augustinus Fonden, https://augustinusfonden.dk/ (HF), Læge Sofus Carl Emil Friis og Hustru Olga Doris Friis' Legat (HF), and A. P. Møller Fonden til Lægevidenskabens Fremme, https://www. apmollerfonde.dk/ansoegning/fonden-til-laegevidenskabens-fremme/ (HF). The funders of the study and the manufacturer of the LNS formulations had no role in study design, data collection and analysis, decision to publish, or preparation of the manuscript.

**Competing interests:** I have read the journal's policy and the authors of this manuscript have the following competing interests: HF and CM have received research grants from Arla Food for Health, HF, BG and CM have received research grants from the Danish Dairy Research Foundation, CM and KFM also received funds from Arla Foods Amba and finally HF, CM, KFM, BG, SF and AB have had research collaboration with Nutriset, a producer of LNS. Other authors declare no financial relationships with any organisations that might have an interest in the submitted work in the previous five years, and declare no other relationships or activities that could appear to have influenced the submitted work

**Abbreviations:** AGP, α1-acid glycoprotein; CI, confidence interval; FFM, fat-free mass; FFMI, fat-free mass index; FM, fat mass; FMI, fat mass index; HAZ, height-for-age z-score; ID, identification number for participants; IGF-1, insulin-like growth factor-1; IQR, interquartile range; KHz, kilohertz; ITT, intention-to-treat; LNS, lipid-based nutrient supplement; MP, milk protein; MUAC, mid-upper arm circumference; SD, standard deviation; UCPH, University of Copenhagen; WASH, water, sanitation, and hygiene; WAZ, weight-for-age z-score; WHO, World Health Organization; WHZ, weight-for-height z-score; WP, whey permeate.

mean (± standard deviation) height-for-age z-score (HAZ) −3.02 ± 0.74 and 12.7% (95) were breastfed. The 750 children were randomized to LNS ($n$ = 600) with or without MP ($n$ = 299 versus $n$ = 301) and WP ($n$ = 301 versus $n$ = 299), or no supplementation ($n$ = 150); 736 (98.1%), evenly distributed between groups, completed 12-week follow-up. Eleven serious adverse events occurred in 10 (1.3%) children, mainly hospitalization with malaria and anemia, all deemed unrelated to the intervention. Unsupplemented children had 0.06 (95% confidence interval, CI [0.02, 0.10]; $p$ = 0.015) decline in HAZ, accompanied by 0.29 (95% CI [0.20, 0.39]; $p$ < 0.001) kg/m$^2$ increase in fat mass index (FMI), but 0.06 (95% CI [−0.002; 0.12]; $p$ = 0.057) kg/m$^2$ decline in fat-free mass index (FFMI). There were no interactions between MP and WP. The main effects of MP were 0.03 (95% CI [−0.10, 0.16]; $p$ = 0.662) cm in height and 0.2 (95% CI [−0.3, 0.7]; $p$ = 0.389) mm in knee-heel length. The main effects of WP were −0.08 (95% CI [−0.21, 0.05]; $p$ = 220) cm and −0.2 (95% CI [−0.7; 0.3]; $p$ = 403) mm, respectively. Interactions were found between WP and breastfeeding with respect to linear growth ($p$ < 0.02), due to positive effects among breastfed and negative effects among non-breastfed children. Overall, LNS resulted in 0.56 (95% CI [0.42, 0.70]; $p$ < 0.001) cm height increase, corresponding to 0.17 (95% CI [0.13, 0.21]; $p$ < 0.001) HAZ increase, and 0.21 (95% CI [0.14, 0.28]; $p$ < 0.001) kg weight increase, of which 76.5% (95% CI [61.9; 91.1]) was fat-free mass. Using height-adjusted indicators, LNS increased FFMI (0.07 kg/m$^2$, 95% CI [0.0001; 0.13]; $p$ = 0.049), but not FMI (0.01 kg/m$^2$, 95% CI [−0.10, 0.12]; $p$ = 0.800). Main limitations were lack of blinding of caregivers and short study duration.

## Conclusions

Adding dairy to LNS has no additional effects on linear growth or body composition in stunted children aged 12 to 59 months. However, supplementation with LNS, irrespective of milk, supports linear catch-up growth and accretion of fat-free mass, but not fat mass. If left untreated, children already on a stunting trajectory gain fat at the expense of fat-free mass, thus nutrition programs to treat such children should be considered.

## Trial registration

ISRCTN13093195

## Author summary

### Why was this study done?

- Stunting affects 149 million children below 5 years and is associated with increased morbidity and mortality, delayed cognitive development and later risk of chronic diseases.

- There are concerns that catch-up growth beyond 2 years may not be possible and that supplementation of already stunted children may lead to fat mass accretion. But no trials have evaluated the effect of lipid-based nutrient supplement (LNS) among children already stunted.

- Milk may support growth and fat-free mass accretion. However, it is expensive and individual effects of milk protein (MP) and whey permeate (WP) are unknown.

- We aimed to assess the effects of MP and WP in large quantity LNS among 1- to 5-year-old children that were already stunted.

### What did the researchers do and find?

- MAGNUS was a randomized 2 × 2 factorial trial with an unsupplemented control group among 750 stunted 1- to 5-year-old children. The effect of large quantity LNS with MP and/or WP on linear growth (primary outcome) and body composition (secondary outcome) was assessed.

- There were no effects of MP or WP on linear growth or body composition. During the 12-week study, height-for-age z-score (HAZ) declined in unsupplemented children, and relative to their height, these children accumulated fat mass and lost fat-free mass.

- In contrast, children receiving LNS—irrespective of milk ingredients—increased in HAZ and fat-free mass, however, not fat mass. There was no difference in catch-up growth between children below and above 2 years.

### What do these findings mean?

- Since children on a growth-faltering trajectory gain fat at the expense of fat-free mass, programs using LNS to treat stunting should be considered.

## Introduction

Globally, 149 million children under 5 years are stunted [1]. Stunting is a multifactorial condition, but inadequate diet quality and recurrent infections are key factors [2]. Stunting is associated with morbidity, mortality, and delayed cognitive development, and later poor educational achievement and working capacity, and risk of chronic diseases [3]. New data suggest that stunting develops interactively with episodes of wasting as a short-term adaptation [4] and increases the risk of death among wasted children [5]. This underscores the need for interventions to prevent or treat stunting.

 Global data show that linear growth faltering starts in utero and that height-for-age z-score (HAZ) declines up to 24 months [6]. This has led to the understanding that interventions to prevent further linear growth faltering, and to support catch-up growth, should be during the first 1,000 days. However, while the decline in the mean global HAZ seems to plateau around 2 years of age, this seems to be due to an increase in the standard deviation (SD) of height with age, and in fact, the deficit in absolute height continues [7]. Moreover, reversal of linear growth faltering and catch-up growth may be possible beyond 2 years of age [8,9]. Yet, while stunting is associated with short- and long-term adverse health outcomes, it may not be stunting per se which has such effects [3]. How much stunting impacts health outcomes directly, and how much it is only a marker of other causes of adverse outcomes, remains poorly understood.

Furthermore, whether nutritional intervention can reduce the functional deficits and risks associated with stunting may not be conditional on catch-up in linear growth.

To date, nutritional interventions have had only modest effects on linear growth. A meta-analysis of lipid-based nutrient supplement (LNS) supplementation of young children in low-income settings showed that these improved by only 0.11 HAZ [10]. The limited effect on linear growth has been partly attributed to poor gut structure and function [11]. However, large trials combining comprehensive water, sanitation, and hygiene (WASH) interventions with small quantity LNS reported no effects from the WASH interventions and only minimal effects from the supplements [12].

The minimal effect of small quantity LNS on linear growth could be due to inadequate amounts of not only energy, but also high-quality protein. To our knowledge, no trial seems to have assessed the effect of large quantity LNS with high-quality protein in the prevention of stunting or of any supplement as a treatment for already stunted children. This may be due to concerns that supplementation of stunted children will lead to excessive accretion of fat mass (FM) rather than fat-free mass (FFM) [13], and therefore, increase the risk of chronic disease later on in adulthood. However, recent studies among children with moderate [14] and severe acute malnutrition [15] showed that even those already stunted predominantly gained FFM when supplemented with LNS. Hence, the concerns may not be justified.

Milk has long been considered important for linear growth [16] and more recently for lean mass [17]. However, milk is expensive, and its inclusion in food supplements increases cost. It is therefore important to document the potential effects of different milk ingredients. Milk proteins (MPs) have a complete amino acid profile and are thought to promote growth by stimulating the growth factors insulin-like growth factor-1 (IGF-1) and insulin [18]. Whey permeate (WP), i.e., a leftover fraction after whey processing containing lactose and bioavailable minerals, may have prebiotic effects as well as a role in lean mass accretion and bone mineralization [19].

We aimed to assess the effects of MP and WP provided as part of LNS, as well as the effect of LNS itself, on linear growth and body composition among stunted children from 12 to 59 months of age.

## Methods

### Study design

We conducted a randomized, double-blinded, community-based trial (MAGNUS) testing the effects of 4 large quantity LNS formulations on linear growth and body composition among stunted children. Of those children enrolled in the study, 600 were randomized to one of the 4 formulations, based on MP or soy protein and WP or maltodextrin using a 2 × 2 factorial design. As a reference, a further 150 children were randomized to no supplement, allowing us to assess the effect of LNS, irrespective of dairy [20].

### Participants

The study was conducted from 2 study sites at Walukuba and Buwenge health centers, Jinja, Eastern Uganda. With support from local village health teams, 12- to 59-month-old children with a HAZ <-2 and weight-for-height z-score (WHZ) ≥-3 were identified in villages surrounding the 2 health centers and referred to a study site for full assessment of eligibility. The age range was chosen to minimize interference with breastfeeding and to assess catch-up growth in children beyond 2 years of age. All children with severe acute malnutrition (mid-upper arm circumference [MUAC] <11.5 cm or WHZ <-3 or bipedal pitting edema) were referred for appropriate treatment at a nearby therapeutic center.

Children were eligible if they were confirmed to have HAZ <-2, and age 12 to 59 months, lived in the catchment area, their caregivers were willing to return for follow-up visits and agreed to phone-follow-up, and home visits. Children with severe acute malnutrition, medical complications requiring hospitalization, a history of peanut or milk allergies, disabilities impeding ability to eat or impeding measurements of height were excluded. Furthermore, children were excluded if they participated in another study. Only 1 child was included per household.

## Randomization, allocation concealment, and blinding

Children were individually randomized to one of the 4 LNS formulations (1:1:1:1) or the reference group (4:1). The LNS sachets were labeled with a unique three-letter code corresponding to the different formulations. Each of the 4 formulations as well as the reference group had 2 unique codes, giving a total of 10 codes used in the allocation sequence list. The study was block-randomized, with variable block sizes of 10 and 20, and stratified by site. Randomization sequences for each study site were computer-generated using R by a staff member at University of Copenhagen (UCPH) who was otherwise not involved in the study. The original was kept in a sealed envelope at UCPH. A hard copy of the site randomization list was provided to the respective site pharmacist in a sealed envelope.

Each enrolled child was sequentially allocated a unique identification (ID) number by an administrative staff member. At the end of baseline activities, in a separate location without staff access, the site pharmacist allocated the intervention or reference according to the hard copy of the random allocation list. Only the pharmacist had access to this allocation list. The pharmacist scanned a QR code to record what was distributed in a spreadsheet, which was submitted to be monitored on a weekly basis by an independent assessor at UCPH.

All investigators and outcome assessors were blinded with respect to LNS versus no LNS and to the ingredients contained in the differently coded LNS sachets. Caregivers were blinded with respect to the type of LNS allocated since the 4 formulations were indistinguishable in terms of appearance, smell, and taste.

## The intervention

All caregivers were offered nutrition counseling at inclusion, in line with the national policy on infant and young child feeding. Those randomized to the intervention arm received 1 LNS sachet of 100 g/day for 12 weeks. The energy and macronutrient contents of the 4 LNS formulations were matched. The 4 formulations of LNS were based on MP or soy protein isolate and WP or maltodextrin. Milk or soy protein constituted 50% of total protein, and a vitamin-mineral-mix was added in addition to the minerals from WP. The experimental and comparative ingredients were decided by the study team, and final LNS formulations were developed, manufactured, and pre-packed by Nutriset (Malaunay, France) in coded zip-lock bags of 14 sachets. Each 100 g LNS sachet provided 530 to 535 kcal that constituted up to half of the average daily energy requirements of a child and satisfied the daily micronutrient requirements, except for vitamin K and niacin (**Table 1**). Those in the reference group received no supplement and so continued with the family diet. The reference group received one 1 kg bar of laundry soap fortnightly. To discourage sharing, caregivers were informed that the LNS was intended for the study child only. When siblings were aged 6 to 59 months, the family received 1 extra ration of the same food supplement at each visit.

Compliance to LNS intake was measured by counting returned empty sachets and asking the caregiver how many sachets the child had consumed since the last visit. Missing sachets or data were all considered as lack of intake.

**Table 1. Composition of large-quantity lipid-based nutrient supplements with milk or soy protein and whey permeate or maltodextrin.**

| | | Milk protein[1] Whey permeate | Milk protein[1] Maltodextrin | Soy protein[2] Whey permeate | Soy protein[2] Maltodextrin |
|---|---|---|---|---|---|
| **Protein quality, DIAAS[3]** | | | | | |
| 6–35 months | | 0.93 | 0.93 | 0.78 | 0.78 |
| 36+ months | | 1.10 | 1.10 | 0.93 | 0.93 |
| Nutrient | **Per 100 g** | | | | |
| **Macronutrients** | | | | | |
| Energy | kcal | 531 | 535 | 530 | 534 |
| Carbohydrates | g | 42 | 43 | 42 | 43 |
| Lactose | g | 15.7 | 0.4 | 15.3 | 0 |
| Proteins | g | 13.9 | 13.5 | 13.9 | 13.5 |
| Milk | g | 7.15 | 6.75 | 0.40 | 0 |
| Vegetable | g | 6.75 | 6.75 | 13.50 | 13.50 |
| Lipids | g | 33.7 | 33.7 | 33.7 | 33.7 |
| Linoleic acid C18:2 | g | 3.0 | 3.0 | 3.0 | 3.0 |
| Linoleic acid C18:3 | g | 0.5 | 0.5 | 0.5 | 0.5 |
| **Minerals** | | | | | |
| Calcium | mg | 691 | 594 | 691 | 594 |
| Copper | mg | 1.65 | 1.65 | 1.65 | 1.65 |
| Iron | mg | 12 | 12 | 12 | 12 |
| Iodine | mg | 127 | 113 | 127 | 113 |
| Magnesium | mg | 199.2 | 175.8 | 199.2 | 175.8 |
| Manganese | mg | 1.8 | 1.8 | 1.8 | 1.8 |
| Phosphorous | mg | 661 | 539 | 661 | 539 |
| Potassium | mg | 1,315 | 985 | 1,315 | 985 |
| Sodium | mg | 84 | 7 | 156 | 79 |
| Selenium | µg | 30 | 30 | 30 | 30 |
| Zinc | mg | 12.5 | 12.5 | 12.5 | 12.5 |
| **Vitamins[4]** | | | | | |
| Vitamin A | µg | 619 | 619 | 619 | 619 |
| Vitamin B1 | mg | 1.2 | 1.1 | 1.2 | 1.1 |
| Vitamin B12 | µg | 3.2 | 3.0 | 3.2 | 3.0 |
| Vitamin B2 | mg | 3.1 | 2.8 | 2.7 | 2.4 |
| Niacin | mg | 14.9 | 14.6 | 14.9 | 14.6 |
| Pantothenic acid | mg | 5.7 | 4.5 | 5.7 | 4.5 |
| Vitamin B6 | mg | 2.1 | 2.0 | 2.1 | 2.0 |
| Biotin | µg | 74.1 | 67.6 | 74.1 | 67.6 |
| Folic acid | µg | 223 | 223 | 223 | 223 |
| Vitamin C | mg | 67.9 | 67.6 | 67.9 | 67.6 |
| Vitamin D | µg | 16.9 | 16.9 | 16.9 | 16.9 |
| Vitamin E | mg | 18 | 18 | 18 | 18 |
| Vitamin K | µg | 30 | 30 | 30 | 30 |

[1] Milk protein isolate (casein and whey).

[2] Soy protein isolate.

[3] Digestible indispensable amino acid score.

[4] Target values by the end of the products' shelf life.

## Study visits

All caregivers returned fortnightly to collect LNS or laundry soap until the 12th week. Distribution was done at the end of each visit by the site pharmacist in a separate location. Caregivers carried LNS or laundry soap in a bag to minimize risk of unblinding investigators or outcome assessors. At baseline, data on the following were collected: demographic information, full clinical examination, WASH assessment in the children's home, minimum dietary diversity score based on 24-h dietary recall including breastfeeding [21], household food insecurity assessment [22], child and maternal anthropometry, and child body composition. Child anthropometrics (weight, height, knee-heel length, triceps, and subscapular skinfolds) and clinical reviews were done at weeks 2, 4, 8, and 12. Body composition data and blood samples were collected at baseline and week 12. Loss to follow-up was defined as those who did not return for the 12-week visit.

## Outcomes

The primary outcomes were height (cm) and knee-heel length (mm), and the primary effect estimate was difference in changes in height and in knee-heel length between the LNS interventions from baseline to week 12. Secondary outcomes included the following: weight (g), body composition by calibrated bioelectrical impedance analysis [FM, FFM, fat mass index (FMI), and fat-free mass index (FFMI) (kg/m$^2$)], skinfold thickness [triceps and subscapular (mm)], MUAC, HAZ, weight-for-age z-score (WAZ), WHZ, and serum IGF-1 (ng/ml). The secondary effect estimates were then differences in changes in these outcomes from baseline to week 12 between intervention groups.

Height (or length for children below 2 years) was measured 3 times using a wooden board (Weigh and Measure, Maryland, United States of America), ensuring 5 points of contact with repositioning between measurements. Knee-heel length was measured using a digital caliper with a 0.01 mm resolution (Mitutoyo, Neuss, Germany) mounted with knee and heel caps cast in hard plastic. The distance between the knee (from the lateral condyle) and the heel (calcaneus) was measured 5 consecutive times on the left leg preferably as the child sat with both legs hanging on the edge of a table or on the caregiver's lap.

Participant and maternal weight were measured using an electronic double weighing scale (SECA 876, Hamburg, Germany) while maternal height was taken using a fixed wall stadiometer (SECA 206, Hamburg, Germany). MUAC and skinfolds were measured using a non-elastic MUAC tape (UNICEF SD, Copenhagen, Denmark) and a Harpenden skinfold caliper (Baty International, West Sussex, England), respectively. Skinfolds thickness were taken on the left side according to the caliper manufacturer's instructions. For site referral and study inclusion, z-scores were calculated using the World Health Organization (WHO) field growth charts while the WHO anthro program (igrowup macro) was used to calculate z-scores in Stata 14 (Stata, College Station, Texas, USA). Body composition was measured in duplicate by bioelectrical impedance using Bodystat 500 machines at 50 kilohertz (KHz) (Bodystat, Isle of Man, United Kingdom). The mean impedance was converted to FFM using an equation developed for this specific population of 1- to 5-year-old Ugandan children with stunting [23]:

$$FFM(kg) = 3.796 + 0.214(HAZ) + 0.488(sex) + 0.068(age) + 0.400\frac{(height)^2}{Z_{50}}.$$

Sex was coded as female = 0 and male = 1, age was in months, height was in cm, and $Z_{50}$ was the mean impedance at 50 KHz. FM was calculated as weight minus FFM. Height-adjusted indices, FMI and FFMI, were calculated as FM or FFM divided by height in meters squared (kg/m$^2$).

Blood samples were collected by venipuncture from the forearm in vacuum tubes. These were stored at ambient temperature, transported to a local lab, left to clot for 30 min, and centrifuged for 10 min at 3,500 rpm (2,300 g). Serum was then transferred to cryotubes and frozen at −20°C. Every week, samples were transported on ice to a biorepository and stored at −80°C (Integrated Biorepository of H3Africa Uganda, Makerere University). Samples were shipped on dry ice to UCPH for IGF-1 analysis and VitminLab (Willstaett, Germany) for α1-acid glycoprotein (AGP) analysis. Serum IGF-1 was analyzed on an Immulite 2000 Analyzer (Siemens Healthcare, GmbH) with intra- and inter-assay CVs of 1.9% to 4.2% and 4.2% to 7.2%, respectively. AGP was analyzed by sandwich enzyme-linked immunosorbent assay with assay variability between 5% and 14%.

## Statistical analyses

To detect a 0.35 SD or greater difference between any 2 groups at 5% significance level and 80% power while allowing for 10% attrition, we recruited 150 children per group (600 in total based on the 4 combinations of MP and WP). If there were no interactions between the 2 experimental arms, then 2 groups of 300 children could be compared enabling differences of 0.24 SD to be detected. In the *Treatfood* trial [14], the SD of height and knee-heel length at baseline was 5.3 cm and 18.1 mm, respectively, so a 0.24 SD difference corresponded to 1.3 cm and 4.3 mm. In secondary analyses, we would assess the effect of LNS (irrespective of milk ingredients) between the 600 (interventional group) and 150 unsupplemented children with the ability to detect a 0.27 SD difference.

Participant data were collected using paper case report forms, double-entered in EpiData (Epidata Association, Odense, Denmark) with inbuilt range checks, and submitted weekly using REDCap (Open Source Vanderbilt University) onto a secure server at UCPH. Statistical analyses were carried out using Stata 14. Descriptive statistics were presented as mean (SD) for normally distributed variables, median (interquartile range) for non-normally distributed continuous variables and frequency % (n) for categorical variables. Distribution was assessed visually using normal probability plots. Mean changes in anthropometry and body composition among unsupplemented children were given with 95% confidence intervals (CIs) and *p* value based on paired *t* tests.

The statistical analyses were based on the intention-to-treat (ITT) principle using available case data. The 2 × 2 factorial design was analyzed using a linear mixed model, where we first tested for any interactions between the 2 interventions in the matrix (MP versus WP). In case of no interaction, we then tested for main effects on linear growth and body composition (i.e., +MP versus -MP and +WP versus -WP). In our secondary analysis, we also compared the intervention, i.e., LNS irrespective of ingredient (600 participants) against the unsupplemented reference group (150 participants). We carried out prespecified subgroup analyses for breastfeeding status, sex, stunting severity, and inflammation (defined as serum AGP, a slow reacting acute phase reactant, >1.2 g/l) to test for any further interactions [20].

All our models included the baseline value of the outcome as fixed effect and participant as random effect. Adjusted analyses further included age, sex, and season as fixed effects and site as random effect to obtain the adjusted measures at 95% CI and *p* < 0.05 significance level. All available data points were included in the models. A per protocol analysis was then carried out including only those children who fulfilled the eligibility criteria and had minimum 80% compliance (supplemented group). For all analyses, model assumptions were assessed visually using residual and QQ plots.

Difference in compliance between LNS formulations, sex, age category, and breastfeeding status were assessed by chi-square tests.

### Ethical considerations

This study was approved by the School of Medicine Research and Ethics Committee of Makerere University, Kampala Uganda (#REC REF2019-013) and the Uganda National Council of Science and Technology (SS4927). Consultative approval was obtained from the Danish National Committee on Biomedical Research Ethics (1906848). The study was registered in the ISRCTN registry (ISRCTN13093195) and conducted in accordance with the principles of the Helsinki Declaration and local guidelines for human research. Oral and written information was provided in Lusoga, Luganda or English. Before caregivers gave written informed consent, their understanding was evaluated by a second staff member, using a questionnaire. An independent data safety and monitoring board conducted interim safety evaluations based on reports of adverse events and reasons for any early termination.

## Results

From 7th of February to 17th September 2020, we screened 7,611 children (**Fig 1**). Of 1,112 stunted children referred to the study sites, 750 (67%) were enrolled and randomized to the 5 groups of which 736 (98.1%) completed the 12-week follow-up. Of the 14 (1.9%) children not seen at the 12-week follow-up, 2 (1.3%) were among the 150 unsupplemented and 12 (2.0%) among the 600 given LNS. Of these, there were no difference by MP (5 versus 7) and WP (6 versus 6). Of 3,750 (750 × 5) potential visits, data were obtained at 3,714 (99.0%).

Eleven serious adverse events were reported among 10 (1.3%) children, 8 (1.3%) of whom received LNS while 2 (1.3%) did not. All were due to hospitalization with life-threatening conditions, mainly severe malaria and anemia. The events were deemed unrelated to the trial intervention and no children died.

At inclusion, the median (interquartile range, IQR) age was 30 (23, 41) months, and the mean ± SD HAZ was −3.02 ± 0.74. The proportion still breastfed was 12.7% (95), 39% (86) among those below and 1.7% (9) among those above 2 years. More than half (59%) had consumed at least 1 animal-source food the previous day. The randomization resulted in baseline equivalence, except for currently breastfed, diverse diet, and positive malaria test (**Table 2**).

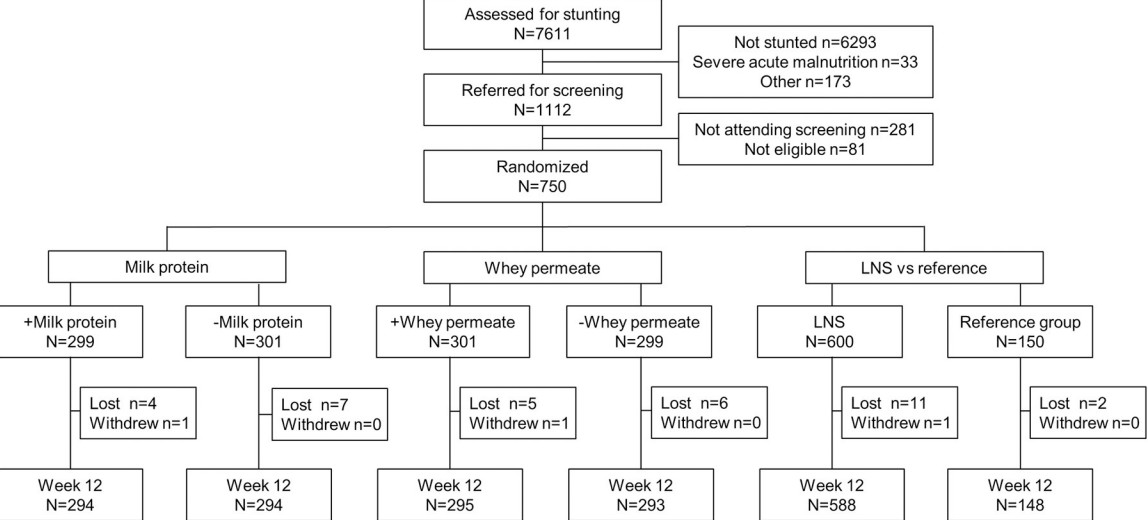

**Fig 1. Trial profile showing the number of children assessed for stunting, screened, recruited to each intervention group and lost to follow-up.** Loss to follow-up was defined as those not returning for the 12-week visit. LNS, lipid-based nutrient supplement.

**Table 2. Baseline characteristics among children with stunting randomized to a lipid-based nutrient supplement (LNS) with milk or soy protein and whey permeate or maltodextrin ($n = 4 \times 150 = 600$), or no LNS ($n = 150$) for 12 weeks[1].**

| | LNS with milk protein ($n = 299$) | LNS with soy protein ($n = 301$) | LNS with whey permeate ($n = 301$) | LNS with maltodextrin ($n = 299$) | LNS ($n = 600$) | No supplement ($n = 150$) |
|---|---|---|---|---|---|---|
| Age (months) | 31 [24,41] | 29 [22,40] | 31 [23,42] | 28 [23,39] | 30 [23,41] | 32 [23,41] |
| Sex (male) | 52.2% (156) | 57.1% (172) | 52.8% (159) | 56.5% (169) | 54.7% (328) | 56.0% (84) |
| Currently breastfed | 14.1% (42) | 12.7% (38) | 16.0% (48) | 10.7% (32) | 13.4% (80) | 10.1% (15) |
| Residence (rural) | 44.5% (133) | 43.5% (131) | 42.9% (129) | 45.2% (135) | 44.0% (264) | 47.3% (71) |
| Household size | 5 [4,7] | 5 [4,6] | 5 [4,7] | 5 [4,6] | 5 [4,7] | 5 [4,7] |
| No maternal education | 48.4% (138) | 47.7% (137) | 50.2% (146) | 45.9% (129) | 48.1% (275) | 44.7% (63) |
| Severe food insecurity[2] | 66.8% (189) | 65.5% (188) | 68.3% (198) | 63.9% (179) | 66.1% (377) | 68.7% (101) |
| Diverse diet[3] | 10.7% (32) | 6.0% (18) | 7.7% (23) | 9.0% (27) | 8.4% (50) | 5.4% (8) |
| **Anthropometric data** | | | | | | |
| Height (cm) | 81.9 ± 7.1 | 81.3 ± 7.5 | 81.9 ± 7.6 | 81.4 ± 7.1 | 81.6 ± 7.3 | 81.9 ± 7.5 |
| Knee-heel length (mm) | 235.4 ± 27.7 | 232.9 ± 29.4 | 235.5 ± 29.1 | 232.8 ± 28.0 | 234.1 ± 28.6 | 235.5 ± 29.6 |
| HAZ (z-score) | −3.02 ± 0.74 | −3.04 ± 0.73 | −3.06 ± 0.75 | −2.99 ± 0.72 | −3.03 ± 0.73 | −2.99 ± 0.75 |
| Weight (kg) | 10.7 ± 1.9 | 10.5 ± 2.0 | 10.6 ± 2.0 | 10.5 ± 1.9 | 10.6 ± 2.0 | 10.6 ± 2.1 |
| FM (kg) | 1.9 ± 0.9 | 1.7 ± 0.8 | 1.8 ± 0.8 | 1.8 ± 0.9 | 1.8 ± 0.9 | 1.8 ± 0.9 |
| FFM (kg) | 8.8 ± 1.4 | 8.8 ± 1.5 | 8.8 ± 1.5 | 8.7 ± 1.4 | 8.8 ± 1.4 | 8.8 ± 1.5 |
| FMI (kg/m$^2$) | 2.7 ± 1.2 | 2.6 ± 1.1 | 2.6 ± 1.1 | 2.7 ± 1.2 | 2.7 ± 1.2 | 2.6 ± 1.2 |
| FFMI (kg/m$^2$) | 13.1 ± 0.7 | 13.1 ± 0.7 | 13.1 ± 0.7 | 13.1 ± 0.7 | 13.1 ± 0.7 | 13.1 ± 0.7 |
| WHZ (z-score) | −0.27 ± 1.03 | −0.42 ± 0.93 | −0.34 ± 0.95 | −0.35 ± 1.01 | −0.35 ± 0.98 | −0.43 ± 1.03 |
| WAZ (z-score) | −1.87 ± 0.91 | −1.97 ± 0.79 | −1.94 ± 0.84 | −1.90 ± 0.87 | −1.92 ± 0.86 | −1.97 ± 0.83 |
| Mid-upper arm circumference (cm) | 14.5 ± 1.1 | 14.4 ± 1.2 | 14.5 ± 1.1 | 14.4 ± 1.2 | 14.4 ± 1.1 | 14.4 ± 1.3 |
| Triceps skinfold (mm) | 8.2 ± 1.7 | 8.1 ± 1.7 | 8.2 ± 1.8 | 8.1 ± 1.6 | 8.1 ± 1.7 | 7.9 ± 1.8 |
| Subscapular skinfold (mm) | 6.8 ±1.8 | 6.7 ±1.7 | 6.8 ±1.8 | 6.7 ±1.7 | 6.7 ±1.7 | 6.6 ±1.7 |
| **Paraclinical data** | | | | | | |
| Malaria (rapid diagnostic test positive) | 39.9% (117) | 36.6% (108) | 40.2% (119) | 36.0% (106) | 38.3% (225) | 45.0% (67) |
| Hemoglobin <110 g/l | 64.9% (192) | 64.7% (192) | 64.3% (191) | 65.2% (193) | 64.8% (384) | 63.3% (95) |
| Serum α$_1$-acid glycoprotein >1.2 g/l | 51.7% (152) | 48.8% (145) | 49.7% (147) | 50.8% (150) | 50.3% (297) | 48.7% (73) |
| Serum insulin-like growth factor-1 (ng/ml) | 43.0 ± 22.8 | 39.2 ± 20.2 | 42.2 ± 21.4 | 40.0 ± 21.8 | 41.1 ± 21.6 | 39.7 ± 20.7 |

[1] Data are median [IQR], % ($n$) and mean ± SD.

[2] Based on household food insecurity access scale.

[3] Based on the minimum dietary diversity score.

FFM, fat-free mass; FFMI, fat-free mass index; FM, fat mass; FMI, fat mass index; HAZ, height-for-age z-score; IQR, interquartile range; LNS, lipid-based nutrient supplement; SD, standard deviation; WAZ, weight-for-age z-score; WHZ, weight-for-height z-score.

## Change in control children

Among the 150 unsupplemented children, there was a 1.68 (95% CI [1.54, 1.82]; $p < 0.001$) cm increase in height during the 12-week study period, corresponding to a 0.06 (95% CI [0.02, 0.10]; $p = 0.015$) decline in HAZ (**Table 3**). Of the 0.58 (95% CI [0.51, 0.64], $p < 0.001$) kg increase in weight, 57.3% (95% CI [53.1, 61.4]) was FFM. To account for the increase in height, the increase in body mass was also indexed for height. Accordingly, there was a 0.29 (95% CI [0.20, 0.39], $p < 0.001$) kg/m$^2$ increase in FMI, but a 0.06 (95% CI [−0.002, 0.12]; $p = 0.057$) kg/m$^2$ decline in FFMI.

## Effects of milk protein and whey permeate

Based on ITT, there was no interaction between MP and WP for any of the outcome variables (**Tables 4 and S1**). Hence, the results of the primary analysis can be presented as the main

**Table 3. Linear growth and body composition during the trial period among the 150 children with stunting randomized to no supplementation for 12 weeks[1].**

| | Baseline | | Endline | | | | |
|---|---|---|---|---|---|---|---|
| | *n* | Mean ± SD | n | Mean ± SD | *n* | Difference (95% CI) | *p* value[2] |
| Height (cm) | 150 | 81.9 ± 7.5 | 148 | 83.6 ± 7.4 | 148 | 1.68 (1.54, 1.82) | <0.001 |
| Knee-heel length (mm) | 149 | 235.5 ± 29.6 | 148 | 242.7 ± 28.8 | 147 | 7.2 (6.6, 7.7) | <0.001 |
| HAZ (z-score) | 150 | −2.99 ± 0.75 | 145 | −3.05 ± 0.79 | 145 | −0.06 (−0.10, −0.02) | 0.015 |
| Weight (kg) | 150 | 10.6 ± 2.1 | 148 | 11.2 ± 2.1 | 148 | 0.58 (0.51, 0.64) | <0.001 |
| FM (kg) | 149 | 1.8 ± 0.9 | 145 | 2.0 ± 1.0 | 144 | 0.25 (0.19, 0.31) | <0.001 |
| FFM (kg) | 149 | 8.8 ± 1.5 | 145 | 9.1 ± 1.5 | 144 | 0.33 (0.29, 0.37) | <0.001 |
| FMI (kg/m²) | 149 | 2.6 ± 1.2 | 145 | 2.9 ± 1.2 | 144 | 0.29 (0.20, 0.39) | <0.001 |
| FFMI (kg/m²) | 149 | 13.1 ± 0.7 | 145 | 13.0 ± 0.7 | 144 | −0.06 (−0.12, 0.002) | 0.057 |
| WHZ (z-score) | 150 | −0.43 ± 1.03 | 145 | −0.20 ± 0.99 | 145 | 0.25 (0.16, 0.33) | <0.001 |
| WAZ (z-score) | 150 | −1.97 ± 0.83 | 145 | −1.87 ± 0.79 | 145 | 0.11 (0.06, 0.17) | <0.001 |
| Mid-upper arm circumference (cm) | 150 | 14.4 ± 1.3 | 148 | 14.6 ± 1.2 | 148 | 0.18 (0.10, 0.27) | <0.001 |
| Triceps skinfold thickness (mm) | 150 | 7.9 ± 1.8 | 148 | 7.9 ± 1.8 | 148 | 0.05 (−0.13, 0.23) | 0.611 |
| Subscapular skin fold thickness (mm) | 150 | 6.6 ± 1.7 | 148 | 6.8 ± 1.8 | 148 | 0.16 (−0.01, 0.33) | 0.081 |
| Serum insulin-like growth factor-1 (ng/ml) | 148 | 39.7 ± 20.7 | 147 | 45.3 ± 20.4 | 145 | 5.5 (3.1, 7.8) | <0.001 |

[1] Data are number, mean ± SD, difference (endline—baseline) with 95% CI and *p* value.

[2] *P* value based on paired *t* test.

CI, confidence interval; FFM, fat-free mass; FFMI, fat-free mass index; FM, fat mass; FMI, fat mass index; HAZ, height-for-age z-score; SD, standard deviation; WAZ, weight-for-age z-score; WHZ, weight-for-height z-score.

**Table 4. Intention-to-treat analysis, adjusted. Effects of milk protein and whey permeate in lipid-based nutrient supplement (LNS) on linear growth and body composition among 750 children with stunting. Primary analysis based on the 2 × 2 factorial design among the 600 children given LNS and secondary analysis based on the 600 children given LNS vs. 150 given no supplement.[1]**

| | | Milk vs. soy protein (*n* = 299 vs. *n* = 301) | | WP vs maltodextrin (*n* = 301 vs. *n* = 299) | | LNS vs. no supplement (*n* = 600 vs. *n* = 150) | |
|---|---|---|---|---|---|---|---|
| | Interaction, *p* value | B (95% CI) | *p* value | B (95% CI) | *p* value | B (95% CI) | *p* value |
| **Primary outcomes** | | | | | | | |
| Height (cm) | 0.478 | 0.03 (−0.10, 0.16) | 0.662 | −0.08 (−0.21, 0.05) | 0.220 | 0.56 (0.42, 0.70) | <0.001 |
| Knee-heel length (mm) | 0.620 | 0.2 (−0.3, 0.7) | 0.389 | −0.2 (−0.7, 0.3) | 0.403 | 1.9 (1.4, 2.4) | <0.001 |
| **Other outcomes** | | | | | | | |
| HAZ (z-score) | 0.568 | 0.01 (−0.03, 0.04) | 0.746 | −0.02 (−0.06, 0.01) | 0.233 | 0.17 (0.13, 0.21) | <0.001 |
| Weight (kg) | 0.114 | 0.06 (−0.01, 0.12) | 0.088 | −0.03 (−0.10, 0.03) | 0.314 | 0.21 (0.14, 0.28) | <0.001 |
| FM (kg) | 0.394 | 0.03 (−0.04, 0.10) | 0.354 | −0.01 (−0.08, 0.05) | 0.678 | 0.04 (−0.03, 0.12) | 0.250 |
| FFM (kg) | 0.402 | 0.03 (−0.01, 0.08) | 0.169 | −0.02 (−0.07, 0.03) | 0.379 | 0.16 (0.11, 0.22) | <0.001 |
| FMI (kg/m²) | 0.717 | 0.04 (−0.06, 0.14) | 0.455 | −0.02 (−0.12, 0.08) | 0.727 | 0.01 (−0.10, 0.12) | 0.800 |
| FFMI (kg/m²) | 0.742 | 0.06 (−0.002, 0.12) | 0.060 | 0.01 (−0.05, 0.07) | 0.730 | 0.07 (0.0001, 0.13) | 0.049 |
| WHZ (z-score) | 0.626 | 0.04 (−0.03, 0.11) | 0.278 | −0.02 (−0.09, 0.05) | 0.572 | 0.08 (0.01, 0.16) | 0.035 |
| WAZ (z-score) | 0.410 | 0.04 (−0.01, 0.09) | 0.129 | −0.03 (−0.08, 0.02) | 0.231 | 0.15 (0.10, 0.21) | <0.001 |
| Mid-upper arm circumference (cm) | 0.546 | 0.06 (−0.01, 0.14) | 0.106 | −0.03 (−0.10, 0.05) | 0.436 | 0.13 (0.05, 0.21) | 0.002 |
| Triceps skinfold thickness (cm) | 0.667 | 0.06 (−0.10, 0.23) | 0.439 | −0.001 (−0.17, 0.16) | 0.990 | 0.08 (−0.10, 0.26) | 0.398 |
| Subscapular skinfold thickness (cm) | 0.951 | 0.03 (−0.13, 0.18) | 0.744 | −0.0002 (−0.16, 0.15) | 0.998 | −0.05 (−0.22, 0.12) | 0.581 |
| Serum insulin-like growth factor-1 (ng/ml) | 0.853 | 2.22 (−0.51, 5.00) | 0.111 | −0.88 (−3.63, 1.85) | 0.526 | 4.18 (1.23, 7.13) | 0.005 |

[1] Data are *p* value for interaction between MP and WP, and main effect B of each intervention with 95% CI and *p* value based on linear-mixed effect models adjusted for age, sex, season, and site.

CI, confidence interval; FFM, fat-free mass; FFMI, fat-free mass index; FM, fat mass; FMI, fat mass index; HAZ, height-for-age z-score; LNS, lipid-based nutrient supplement; WAZ, weight-for-age z-score; WHZ, weight-for-height z-score; WP, whey permeate.

effect for each of the experimental ingredients. There were no effects of either MP or WP in LNS on the primary outcomes: milk compared to soy protein was associated with a 0.03 (95% CI [−0.10, 0.16]; $p = 0.662$) cm increase in height and a 0.2 (95% CI [−0.3, 0.7]; $p = 0.389$) mm increase in knee-heel length. For WP versus maltodextrin, the corresponding estimates were −0.08 (95% CI [−0.21, 0.05]; $p = 0.220$) cm and −0.20 (95% CI [−0.67, 0.27], $p = 0.403$) mm. MP was associated with 0.06 (95% CI [−0.002, 0.12]) kg/m$^2$ greater increase in FFMI, but this was not significant ($p = 0.060$). There were no effects on other anthropometric outcomes, body composition, or on serum IGF-1. The estimates were similar without adjustments for age, sex, and season (**S1 Table**), and with additional adjustments for current breastfeeding, dietary diversity, and positive malaria test.

We assessed if the effects of MP and WP were modified by sex, breastfeeding status, stunting severity, and inflammation at baseline. There was an interaction between MP and inflammation with respect to height ($p = 0.024$) corresponding to a 0.30 (95% CI 0.04, 0.56) cm larger increase in children with compared to without inflammation at inclusion. Among children with inflammation at baseline, MP was associated with a 0.20 (95% CI [0.01, 0.38]; $p = 0.036$) cm larger increase in height. There was no effect modification by sex, breastfeeding, or stunting severity with respect to the primary outcomes. The effects of MP on weight and MUAC were smaller among children with severe stunting (interaction $p = 0.035$ and $p = 0.019$, respectively). Accordingly, in children with moderate stunting, MP was associated with 0.12 (95% CI [0.03, 0.20]; $p = 0.008$) kg larger weight gain and 0.14 (95% CI [0.04, 0.24]; $p = 0.005$) cm larger MUAC gain (**S2 Table**).

The effects of WP on both primary outcomes were modified by breastfeeding (**S3 Table**). Interestingly, this was due to positive effects of WP on linear growth among breastfed, but negative effects among non-breastfed children. As such, WP increased height by 0.44 (95% CI [0.08, 0.81]; $p = 0.017$) cm in breastfed but reduced it by 0.16 (95% CI [0.02, 0.30]; $p = 0.025$) cm in non-breastfed children (interaction $p = 0.003$). Similarly, WP increased knee-heel length by 1.3 (95% CI [0.01, 2.7]; $p = 0.049$) mm in breastfed children, but not among non-breastfed (−0.4 mm, 95% CI [−0.9, 0.1]; $p = 0.095$, interaction, $p = 0.016$). On the other hand, there was a 0.18 (95% CI [0.01, 0.35]; $p = 0.022$) kg/m$^2$ decrease in FFMI by WP among the breastfed, but not in non-breastfed (0.04 kg/m$^2$, 95% CI [−0.02, 0.11]; $p = 0.290$; interaction $p = 0.018$) children. We performed sensitivity analysis to assess if the interaction with breastfeeding was due to age. There was no interaction by age above or below 2 years either with or without the children who were breastfed. Moreover, when we tested for interaction by breastfeeding among children below 2 years, we found similar interactions and effect sizes.

## Effects of LNS

In the secondary analysis, LNS, irrespective of milk ingredients, resulted in a 0.56 (95% CI [0.42, 0.70]; $p < 0.001$) cm increase in height and 1.9 (95% CI [1.4, 2.4], $p < 0.001$) mm increase in knee-heel length (**Table 4**). The effect on height corresponded to a 0.17 (95% CI [0.13, 0.21]; $p < 0.001$) increase in HAZ, which was not different in children below and above 2 years of age (0.19 versus 0.16, $p = 0.555$). LNS resulted in 0.21 (95% CI [0.14, 0.28]; $p < 0.001$) kg larger weight gain, accompanied by 0.08 (95% CI [0.01, 0.16], $p = 0.035$) increase in WHZ. The 0.21 kg increase in weight was driven by FFM gain (0.16 kg, 95% CI [0.11, 0.22]; $p < 0.001$) rather than FM (0.04 kg, 95% CI [−0.03, 0.12]; $p = 0.250$). Hence, FFM comprised 76.5% (95% CI [61.9, 91.1]) of the weight gained due to LNS supplementation. When expressed as height-adjusted indices, the effect of LNS was largely due to an increase in FFMI (0.07 kg/m$^2$, 95% CI [0.0001, 0.13]; $p = 0.049$), rather than FMI (0.01 kg/m$^2$, 95% CI [−0.10, 0.12]; $p = 0.800$). LNS also resulted in a 4.18 (95% CI [1.23, 7.13], $p = 0.005$) ng/ml increase in

serum IGF-1. The estimates were similar without adjustments for age, sex, and season (S1 Table) and with additional adjustments for current breastfeeding, dietary diversity, and positive malaria test. LNS subgroup analyses are described in **S4 Table**.

Among the 600 children receiving LNS, 86% of the caregivers returned more than 80% empty sachets. There was no difference in compliance by LNS formulation or sex ($p > 0.466$). Compliance was lower among children below compared to above 2 years of age (80% versus 88%, $p = 0.013$), but among children below 2 years, there was no difference by breastfeeding (80% versus 80%). The per protocol analyses showed similar results as ITT (**S5 Table**)

## Discussion

This study showed that whether an LNS supplement contained MP or WP had no impact on increment in height, but MP tended to increase FFMI. However, compared to unsupplemented children, we identified clear benefits of LNS supplementation for both growth and tissue accretion. Whereas unsupplemented children continued to become more stunted while also gaining fat, those supplemented demonstrated 0.17 z-score recovery in height and gained FFM but not FM.

Previous studies indicate that milk may improve linear growth [16,24] and maybe more so in low-income settings [16]. Accordingly, foods for treatment of severely malnourished children are currently required to contain minimum 50% of total protein from milk. However, we found no main effects of MP or WP in LNS on linear or ponderal growth outcomes among stunted children. A trend was found for an effect of MP on FFMI. Recent studies investigating the effect of milk in LNS on prevention of stunting [25,26] or treatment of moderate acute malnutrition also found no or limited effects [14,27], but a meta-analysis showed an effect of a high milk content in LNS on weight gain among severely malnourished children [28]. Another meta-analysis on milk intake, which included only data from randomized trials and mainly from high-income settings, also did not find any effect of dairy intake on linear growth, but a small effect on lean mass, in 6- to 18-year-old children [17]. Although there were no main effects of MP or WP in the current study, it is noticeable that all effect sizes were positive for MP and negative for WP. If this could be confirmed in other trials, LNS with milk ingredients containing MP but not WP or lactose would be recommended.

Multiple factors may explain the inconsistent findings in studies assessing the effect of milk or MP on growth and body composition. In the current study, MP was compared against soy protein isolate, i.e., a protein with a high protein quality score and low content of antinutrients [29,30]. This shows the importance of considering the control intervention when interpreting our findings and could indicate that differences in linear growth observed in previous studies [16,24] may have been due to differences in protein quantity, quality, or bioavailability more than a specific milk factor. The amount of LNS or milk ingredients in food supplements as well as the degree of malnutrition likely also affect the results. Studies among 6-month-old infants from Malawi showed no overall effect of small or medium quantity milk-based LNS on development of stunting or change in LAZ over 12 months [25,26]. In another Malawian study, moderately wasted children received approximately 75 kcal/kg/day (approximately 100 g/day) of LNS with whey protein and whey permeate versus dehulled soy-based LNS. The study found small improvements in recovery and ponderal growth outcomes but no effect on linear growth [27]. In severely malnourished children where children's diets are completely replaced, the meta-analysis mentioned above [28] found effect of LNS containing 50% versus low or no protein from milk on weight but not height.

We found that the effect of MP on height and HAZ, but not knee-heel length was stronger in children with inflammation at baseline. While frequent infections and inflammation are

known to reduce growth, possibly via down-regulation of IGF-1 [31], our finding was not explained by a similar interaction with respect to IGF-1.

Interestingly, we found very strong interactions between WP and breastfeeding for all measures of linear growth. WP increased linear growth in breastfed children, but reduced it in non-breastfed children. A possible explanation could be the high lactose content of WP (approximately 15 g/day). The gut microbiota of breastfed children have significantly more lactobacilli and bifidobacteria [32], which can ferment lactose to short chain fatty acids. In contrast, undigested lactose reaching the large intestine of non-breastfed children may result in energy loss or lead to osmotic diarrhea and thereby reduced growth [33,34].

Compared to the unsupplemented children who were on a growth-faltering trajectory, LNS resulted in a 0.17 HAZ increase. This effect size is rather high compared to what has been reported in other studies. Meta-analyses evaluating the effect of complementary feeding interventions, LNS or small quantity LNS on linear growth in 6 to 23 months old children in different food-insecure or low-income settings found increases of 0.08, 0.11, and 0.14 LAZ, respectively, compared to unsupplemented children [10,35,36]. The larger effect of LNS on linear growth in the current study may be due to a higher amount of energy and high-quality protein in large quantity LNS (100 g/day) compared to small (10 to 20 g/day) or medium quantity LNS (40 to 50 g/day) used in previous studies.

Children receiving LNS gained on average 0.16 kg more FFM. The height-adjusted index, FFMI, also increased slightly more in LNS supplemented children, which indicates that the FFM increase is not only in proportion to increase in height, but also catch-up of FFM. In contrast, FM and FMI were not affected by LNS supplementation. Skinfold data corroborated an absence of effect of LNS on fat mass. The unsupplemented group, nevertheless, increased in FMI and tended to decline in FFMI. This unhealthy change in body composition is very different from well-nourished children from the UK, where FMI centiles decreased and FFMI centiles slowly increased between 1 and 5 years [37]. Studies in both Asia and Africa have shown that poor early growth may be accompanied by a relative preservation of FM alongside reductions in height and FFM [38]. This may be attributed to the survival value of body fat in malnourished young children, demonstrated by a link of low leptin with mortality [39]. A trial among moderately malnourished children also found considerable improvement of FFM and FFMI accretion in 6- to 23-month-old children receiving LNS versus corn-soy blend [14].

This trial may be the first to investigate the effect of large quantity LNS on already stunted children. The main strengths are the randomized 2 × 2 factorial design with an unsupplemented control group, a high follow-up rate, and inclusion of body composition data. Limitations include lack of blinding of caregivers in regards to LNS supplementation versus no supplementation, but any change in behavior caused by this is likely to cause underestimation of the effects of LNS. There is also a risk that outcome assessors may have been unblinded to LNS supplementation in a few cases, despite all efforts to avoid it. This knowledge might have affected assessment of linear growth, but not body composition as this was measured in an automated procedure. Other limitations are the relatively short duration of the intervention and lack of long-term follow-up.

There were no effects of MP isolate versus soy protein isolate or WP versus maltodextrin on linear growth or body composition among stunted children. However, in the context of continuous growth faltering, supplementation with LNS resulted in a 0.17 HAZ catch-up growth, considerable FFM accretion, an increase in FFMI, and no effects on FM or FMI. This contrasts with unsupplemented children on a stunting trajectory, who are gaining fat at the expense of FFM. Supplementation with large quantity LNS, irrespective of milk content, prevents further linear growth faltering and supports catch-up growth with accretion of fat-free tissue, but not fat. Programs using LNS to treat stunting should be considered.

## Supporting information

**S1 Table. Intention-to-treat analysis.** Effects of milk protein and whey permeate in lipid-based nutrient supplement (LNS) on linear growth and body composition among 750 children with stunting. Unadjusted analyses.
(DOCX)

**S2 Table. Subgroup effects of milk protein in lipid-based nutrient supplement (LNS) on growth by sex, breastfeeding status, stunting severity, and inflammation among children with stunting who received LNS.** Adjusted and unadjusted analyses.
(DOCX)

**S3 Table. Subgroup effects of whey permeate in lipid-based nutrient supplement (LNS) on growth by sex, breastfeeding status, stunting severity, and inflammation among children with stunting who received LNS.** Adjusted and unadjusted analyses.
(DOCX)

**S4 Table. Subgroup effects of lipid-based nutrient supplement (LNS) on growth by sex, breastfeeding status, stunting severity, and inflammation among 750 children with stunting who received LNS ($n = 600$) or no supplement ($n = 150$).** Adjusted and unadjusted analyses.
(DOCX)

**S5 Table. Per protocol analyses.** Effects of milk protein and whey permeate in lipid-based nutrient supplement (LNS) on linear growth and body composition among 750 children with stunting. Adjusted analyses.
(DOCX)

**S1 Text. MAGNUS study protocol.**
(DOCX)

**S2 Text. CONSORT checklist.**
(DOC)

**S3 Text. Statistical analysis plan.**
(PDF)

## Author Contributions

**Conceptualization:** Mette F. Olsen, André Briend, Kim F. Michaelsen, Christian Mølgaard, Ezekiel Mupere, Henrik Friis, Benedikte Grenov.

**Data curation:** Joseph Mbabazi, Henrik Friis, Benedikte Grenov.

**Formal analysis:** Joseph Mbabazi, Christian Ritz, Henrik Friis.

**Funding acquisition:** Henrik Friis.

**Investigation:** Joseph Mbabazi, Hannah Pesu, Rolland Mutumba, Jack I. Lewis, Ezekiel Mupere, Benedikte Grenov.

**Methodology:** Jonathan C. Wells.

**Project administration:** Ezekiel Mupere, Henrik Friis, Benedikte Grenov.

**Supervision:** Joseph Mbabazi, Rolland Mutumba, Ezekiel Mupere, Henrik Friis, Benedikte Grenov.

**Validation:** Joseph Mbabazi, Rolland Mutumba, Benedikte Grenov.

**Writing – original draft:** Joseph Mbabazi.

**Writing – review & editing:** Joseph Mbabazi, Hannah Pesu, Rolland Mutumba, Suzanne Filteau, Jack I. Lewis, Jonathan C. Wells, Mette F. Olsen, André Briend, Kim F. Michaelsen, Christian Mølgaard, Christian Ritz, Nicolette Nabukeera-Barungi, Ezekiel Mupere, Henrik Friis, Benedikte Grenov.

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
