## [Editor Report · Decision Letter 0]

29 Dec 2022

Dear Dr Grenov, 

Thank you for submitting your manuscript entitled "Effect of milk protein and whey permeate in large-quantity lipid-based nutrient supplement on linear growth and body composition among children with stunting: a randomised 2x2 factorial trial in Uganda" for consideration by PLOS Medicine.

Your manuscript has now been evaluated by the PLOS Medicine editorial staff and I am writing to let you know that we would like to send your submission out for external peer review.

Please re-submit your manuscript within two working days, i.e. by Jan 02 2023 11:59PM.

Kind regards,

Philippa Dodd, MBBS MRCP PhD

PLOS Medicine

---

## [Decision Letter · Decision Letter 1]

31 Jan 2023

Dear Dr. Grenov,

Thank you very much for submitting your manuscript "Effect of milk protein and whey permeate in large-quantity lipid-based nutrient supplement on linear growth and body composition among children with stunting: a randomised 2x2 factorial trial in Uganda" (PMEDICINE-D-22-03930R1) for consideration at PLOS Medicine. 

[LINK]

In light of these reviews, I am afraid that we will not be able to accept the manuscript for publication in the journal in its current form, but we would like to consider a revised version that addresses the reviewers' and editors' comments. Obviously we cannot make any decision about publication until we have seen the revised manuscript and your response, and we plan to seek re-review by one or more of the reviewers. 

We expect to receive your revised manuscript by Feb 21 2023 11:59PM. Please email us (plosmedicine@plos.org) if you have any questions or concerns.

We look forward to receiving your revised manuscript. 

Sincerely,

Philippa Dodd, MBBS MRCP PhD

PLOS Medicine

plosmedicine.org

GENERAL

Please respond to all editor and reviewer comments detailed below, in full.

Please complete the CONSORT checklist and ensure that all components of CONSORT are present in the manuscript, including [how randomization was performed, allocation concealment, blinding of intervention, definition of lost to follow-up, power statement]. 

When completing the checklist, please use section and paragraph numbers, rather than page numbers as these often change in the event of publication.

COMMENTS FROM THE ACADEMIC EDITOR

The trial is interesting and looks solid, notwithstanding industry funding, the authors have been fair and transparent in their findings

One question is why this trial was done in older infants (mean age (30 months)?

Further, is 12 weeks a reasonable follow up given all the questions about sustained benefits? 

ABSTRACT

Please report your abstract according to CONSORT for abstracts, following the PLOS Medicine abstract structure (Background, Methods and Findings, Conclusions, as below)

http://www.consort-statement.org/extensions?ContentWidgetId=562

Please structure your abstract using the PLOS Medicine headings (Background, Methods and Findings, Conclusions).

Please combine the Methods and Findings sections into one section, “Methods and findings”.

Abstract Background: It would be helpful to elaborate on some of the context here. What do we know about the different supplements, why these specifically? Why is the comparison of these three supplements an important one to make? For example

Abstract Methods and Findings:

Line 7: Please revise the sub-heading to “Methods and Findings” (as above). Suggest “We performed a randomized…” or something similar

Please include further details of population/setting rural Vs urban for example. 

Any further details on the community-based nature of the study? Were community clinics involved? What kind? For example – this was done rather well in the methods section of the main manuscript

Please specify who was blinded to the intervention and control.

Please provide the number of participants lost to follow up in each group.

Please define all abbreviations at first use including those for statistical reporting IQR, SD, CI, for example and those used for weights and heights.

Please quantify the main results with p values, as well as 95% CIs.

When reporting p values please report as p<0.001 (not .001) or where higher as p=0.002 (not .002), for example

Suggest revising statistical reporting, including throughout the main manuscript, to read as follows:

“0.06 (95%CI [0.02,0.10]; p<0.001)” note the addition of parentheses around CIs and the altered use of semi-colons/colons

Please ensure that all numbers presented in the abstract are present and identical to numbers presented in the main manuscript text.

Please include any variable factors that were adjusted for in your analyses.

Please include a summary of adverse events that you refer to.

In the last sentence of the Abstract Methods and Findings section, please describe the main limitation(s) of the study's methodology.

Line 23-25 – suggest revising this paragraph for improved clarity, especially for the general reader.

Line 146: “…above and below 2 years, children aged 12-59 months…” but here (in the abstract) you refer to the median/IQR age range only. Suggest additional details here for clarity to prevent confusion perhaps?

Abstract Conclusions:

Please interpret the study based on the results presented in the abstract, emphasizing what is new without overstating your conclusions.

* Please avoid vague statements such as "these results have major implications for policy/clinical care". Mention only specific implications substantiated by the results

AUTHOR SUMMARY

Thank you for including an author summary which requires revision. The author summary should follow a similar structure as the abstract (and main manuscript) introducing the need for the study, the details of the study and its findings and finally the study outcomes and how these may impact the bigger picture. 

Please see below for further guidance. 

https://journals.plos.org/plosmedicine/s/revising-your-manuscript#loc-author-summary

It may also help you to review some examples from recently published manuscripts which you can find on our homepage here: https://journals.plos.org/plosmedicine/

METHODS and RESULTS

Please move the ethics statement to the end of the methods section of the main manuscript 

There are a lot of abbreviations used throughout the manuscript, please consider re-defining some of these at first use in the methods section for example where you describe the study design at line 136 onwards. 

Please also ensure that all abbreviations introduced later in the manuscripts are also clearly defined – ID (line 168) and KHz (line 231), SD (line 251) for example. Please check carefully throughout and amend where necessary

Please include the study protocol document and analysis plan, with any amendments, as Supporting Information to be published with the manuscript if accepted.

Please specific in the main methods section that analysis is intention to treat (as for the abstract)

Line 251: “To detect a 0.35 SD…” please define SD for the reader, please also see comments below regarding tables.

TABLES

Please ensure all tables are associated with an appropriate caption/legend which clearly describes their content without the need to refer to the manuscript text. Please ensure that any and all abbreviations including those used for statistical reporting are defined within the caption/legend

Throughout, please replace semicolons, used to differentiate upper and lower bounds, with commas

Table 2: baseline characteristic reporting utilises a variety of different units but the footnote only helps partially because it does not clearly detail what applies where – please revise accordingly – I note superscript text delineating the units used rather inconsistently placed and not clearly defined in the footnote.

Tables 3 and 4 – please indicate whether your analyses are adjusted or unadjusted and in the event of adjustments please state in the captions what factors are adjusted for. In addition, for the purpose of transparent data reporting please also include the unadjusted analyses for comparison.

SUPPLEMENTARY TABLES

Please ensure that each has an appropriate caption which clearly details the table contents without the need to refer to the text

Please revise the presentation of data throughout. It is not clear to the reader what the numerical values within the column between p values and 95% CIs represent 

Please replace semicolons with commas when reporting upper and lower bounds

In places there are closing parentheses in the absence of opening parentheses, please include both. Suggest combining these columns for each sub-group, re-labelling the column header and reporting as follows – for example, row 1, columns 2 and 3 should be combined to a single column and read: -0.21 (-0.47, 0.05) 

Please check and revise throughout all the tables

Throughout, please indicate whether analyses are adjusted or unadjusted and detail which factors are adjusted for in the caption/legends. As above, where adjusted analyses are reported please also include unadjusted analyses for comparison.

FIGURES

Please ensure all figures are associated with an appropriate caption/legend which clearly describes their content without the need to refer to the manuscript text. Please ensure that any and all abbreviations including those used for statistical reporting are defined within the caption/legend

Figure 1- suggest revising the title to include details of what the flow chart shows

DISCUSSION

Please remove all sub-headings from the discussion such that the discussion reads as a single piece of continuous prose and please structure as follows: a short, clear summary of the article's findings; what the study adds to existing research and where and why the results may differ from previous research; strengths and limitations of the study; implications and next steps for research, clinical practice, and/or public policy; one-paragraph conclusion. 

Please remove funding/competing interests/data statements from the end of the discussion and include only in the manuscript submission form in the relevant sections. In the event of publication these will be compiled as metadata.

REFERENCES

Please ensure that citations are placed within square parentheses which precede punctuation 

For example, line 75 should read: “Globally, 149 million children under 5 years are stunted [1].” Please check and revise throughout

Comments from the reviewers:

Reviewer #1: This randomised, double-blind, community-based 2x2 factorial trial aims to assess the effects of MP and WP in lipid-based nutrient supplement (LNS), and of LNS itself, on linear growth and body composition among children with stunting.

Comments:

Can the authors please provide copies of the study protocol, pre-specified statistical analysis plan, and associated guideline checklist for this study?

"Children were individually randomised to one of the four LNS formulations (1:1:1:1) or the reference group (4:1)."

and

"All investigators and outcome assessors were blinded with respect to LNS vs no LNS and to the ingredients contained in the differently coded LNS sachets. Caregivers were blinded with respect to the type of LNS allocated since the four formulations were indistinguishable in terms of appearance, smell and taste. "

The authors have followed a rigorous study design, helping to minimise potential sources of bias.

"To detect a 0.35 SD or greater difference between any two groups at 5% significance level and 80% power while allowing for 10% attrition, we recruited 150 children per group (600 in total based on the 4 combinations of MP and WP). If there were no interactions between the two experimental arms, then two groups of 300 children could be compared enabling differences of 0.24 SD to be detected."

The authors have suitably provided the basis for the sample size calculation, with associated assessment of study power.

"Descriptive statistics were presented as mean (SD), median (interquartile range) for continuous variables and frequency % (n) for categorical variables. Mean changes in anthropometry and body composition among unsupplemented children were given with 95% confidence intervals and p value based on paired t-tests. The statistical analyses were based on the intention-to-treat (ITT) principle using available case data. The 2x2 factorial design was analysed using a linear mixed model, where we first tested for any interactions between the two interventions in the matrix (MP vs WP). In case of no interaction, we then tested for main effects on linear growth and body composition (i.e. +MP vs -MP and +WP vs -WP). In our secondary analysis, we also compared the intervention i.e. LNS irrespective of ingredient (600 participants) against the unsupplemented reference group (150 participants). We carried out prespecified subgroup analyses for breastfeeding status, sex, stunting severity, and inflammation (defined as serum AGP>1 g/l) to test for any further interactions."

The authors have applied technically appropriate statistical methods within the context of this research.

Can the authors please clarify if and how they assessed distributional assumptions (e.g. normality and linearity) in the choice of application of statistical methods?

"All our analyses were adjusted for the baseline measure of the outcome, age, sex, and season as fixed effects, and site plus participant as random effects to obtain the adjusted measures at 95% confidence interval and p<0.05 significance level."

The authors have suitably adjusted the analyses for covariates, including the stratification variable site.

"A per protocol analysis was then carried out including only those children who fulfilled the eligibility criteria and had minimum 80% compliance (supplemented group). Difference in compliance between LNS formulations, sex, age category, and breastfeeding status were assessed by chi-square tests."

The authors have also satisfactorily undertaken additional sensitivity and exploratory analyses, helping to demonstrate the robustness of the study findings.

"Of 1193 stunted children referred to the study sites, 750 (63%) were enrolled and randomised to the five groups of which 736 (98.1%) completed the 12-week follow-up."

Can the authors please comment on whether the included cohort can be considered to be representative, for generalisability of the study findings?

"The randomisation resulted in baseline equivalence with respect to both the primary and secondary comparisons (Table 2)."

Can the authors please check if 'currently breastfed', 'diverse diet', and 'Malaria (RDT positive)' are sufficently balanced between groups, in particular between LNS and no supplement?

"models adjusted for baseline value, age, sex, season, site and ID2"

Whilst it is noted that the authors have conducted a subgroup analysis by breastfeeding, did the authors consider adjusting for the above named variables in the models as a sensitivity analysis?

"Limitations include incomplete blinding of caregivers in regards to LNS supplementation."

Can the authors please expand on the discussion of the main study limitations?

Overall, this is a well written article, with accurately presented Results.

Reviewer #2: Thank you for this opportunity to review this well-written manuscript and preview the results from this trial. Some comments and thoughts below for your consideration: 

1. The tables can be improved and be made stand alone by adding some footnotes such as definitions for terms like severe food insecurity and diet diversity in the main text. Same for supplementary tables. Supplementary Table 4 is missing a column of p-value for interaction between MP and WP. 

2. This may be a minor point - classically, a 2X2 trial assigns the first intervention and then randomizes within the two groups to the second intervention to result in four groups. The statistical analysis are appropriate but the methods indicate that the children were assigned to 5 separate arms and it will be good to be consistent or describe how the randomization sequence was operationalized to address this. 

3. I am also not sure re: the double blinded claim here since the control group and the outcome assessors knew they were not receiving any supplement? Perhaps elaborate in limitations - line #431. 

4. * Lines #251 and #255 - Please mention the significance or rational behind using 0.35 SD and 0.24 SD (or provide appropriate reference) 

* Line #264 For continuous variables, were the data distributed normally? 

5. Line #295 - Data obtained according to age division (below/above 2y) not present in any of the tables. 

Reviewer #3: I have read with great interest the manuscript by Mbabazi and colleagues on the effects of milk and whey protein in large quantity LNS supplements on height and body composition. The research group behind the research is well known for very well designed and executed studies, and this study is no exception. 

The authors report that neither the addition of milk protein (MP) or whey protein (WP) had an additional effect on height gains in stunted children in Uganda. This finding is important, as milk is an expensive ingredient in ready-to-use supplements, and leaving out milk could potentially reduce cost, thereby enabling more children to be treated for the same budget. 

The manuscript is well written and the results and conclusions are well presented. My main concern with the manuscript as it stands now is some discussion on the sample size. The authors were in a sense lucky in that there were no statistical interactions between MP and WP, enabling them to combine the groups, giving them approximately 300 children per group (MP vs no MP ; WP vs no WP). Looking at supplementary table 4, what strikes me is that for the MP vs no MP comparison, all anthropometric indicators are positive, while for the WP vs no WP comparisons, all of them (except FFMI) are negative. Even though non of them are statistically significant, the question arises whether this would also have been the case if the sample size would have been twice as large. The main conclusion of the paper, no impact of MP on height, would probably not have changed, but outcomes such as FFMI or WHZ might have become significant, as well as perhaps a negative effect of WP on HAZ. Hence, a paragraph in the discussion section on the sample size would be welcome.

Minor concerns

Review plos med

Abstract.

Lines 23-26. None of the effects of milk protein or whey protein on height or knee-heel length are statistically significant. I'm fine with the way these effects are presented. But I don't understand why then the effects of MP on FFMI need to be specified as 'non-significant'. Perhaps start this paragraph with stating that none of the effects of MP or WP were statistically significant. And then the specific mentioning of the effect of MP on FFMI can be omitted.

Line s31-32. A 0.21 kg increase in weight corresponding to an increase in how many Z-score (WAZ). And how many WHZ/WLZ scores?

Line 116 low income (2 words)

Line 121-122. While this hypothesis is valid in itself, most of the SQ-LNS trials reported in the paragraph above had cow milk as main protein source.

Line 126 increases

Line 193. I'm fine with the randomisation of the 4 different LNS types, but how can you blind the study team to the 'no-LNS' group, as they will go home without LNS sachets

Line 276 secondary analyses or In a secondary analysis

Line 294. Any justification for using AGP only to indicate systemic inflammation?

[LINK]

---

## [Decision Letter · Decision Letter 2]

21 Mar 2023

Dear Dr. Grenov,

Thank you very much for re-submitting your manuscript "Effect of milk protein and whey permeate in large quantity lipid-based nutrient supplement on linear growth and body composition among stunted children: a randomized 2x2 factorial trial in Uganda" (PMEDICINE-D-22-03930R2) for review by PLOS Medicine.

I have discussed the paper with my colleagues and the academic editor and it was also seen again by 2 reviewers. I am pleased to say that provided the remaining editorial and production issues are dealt with we are planning to accept the paper for publication in the journal.

[LINK]

We look forward to receiving the revised manuscript by Mar 28 2023 11:59PM.   

Sincerely,

Philippa Dodd, MBBS MRCP PhD

PLOS Medicine

plosmedicine.org

Requests from Editors:

GENERAL

Thank you for your very detailed and considered responses to previous editor and reviewer comments. Please see below for further comments from the editorial team.

INDUSTRY FUNDING

Thank you for your clear response to this question. Please include the following “Arla Food for Health, is a public-private research partnership between the University of Copenhagen, Aarhus University and the dairy company, Arla.” Or similar, in the relevant section of the manuscript submission form.

AUTHOR SUMMARY

Thank you for modifying the summary which reads very nicely but is a little too long – suggest the following revisions 

“Why was this study done”:

• Stunting affects 149 million children below 5 years and is associated with increased morbidity and mortality, delayed cognitive development and later risk of chronic diseases.

• There are concerns that catch-up growth beyond 2 years may not be possible and that supplementation of already stunted children may lead to fat mass accretion. But no trials have evaluated the effect of lipid-based nutrient supplements (LNS) among children already stunted. 

• Milk may support growth and fat-free mass accretion. However, it is expensive and individual effects of milk protein and whey permeate are unknown. 

• We aimed to assess the effects of milk protein and whey permeate in large quantity LNS among 1-5 year-old children that were already stunted.

“What did the researchers do and find”

• Line 68 – suggest a separate bullet point for sentence beginning “In contrast…”

SOCIAL MEDIA

To help us extend the reach of your research, please provide any Twitter handle(s) that would be appropriate to tag, including your own, your co-authors’, your institution, funder, or lab. 

Please detail any handles you wish to be included when we tweet this paper, in the manuscript submission form when you re-submit the manuscript.

Comments from Reviewers:

Reviewer #1: Many thanks to the authors for satisfactorily considering and responding to each comment in turn, undertaking additional sensitivity analyses and amending the reporting of the study accordingly.

Reviewer #2: The responses appear to adequately address the concerns. However, the manuscript file sent for review was the previous version without the changes. Thanks.

[LINK]

---

## [Editor Report · Decision Letter 3]

29 Mar 2023

Dear Dr Grenov, 

On behalf of my colleagues and the Academic Editor, Professor Zulfiqar Bhutta, I am pleased to inform you that we have agreed to publish your manuscript "Effect of milk protein and whey permeate in large quantity lipid-based nutrient supplement on linear growth and body composition among stunted children: a randomized 2x2 factorial trial in Uganda" (PMEDICINE-D-22-03930R3) in PLOS Medicine.

PRESS

Best wishes,

Pippa 

Philippa Dodd, MBBS MRCP PhD 

PLOS Medicine